

# Taxonomic revision of *Dasydorylas* Skevington, 2001 (Diptera, Pipunculidae) in the Middle East

Behnam Motamedinia[1,2], Jeffrey H. Skevington[1,3] and Scott Kelso[1]

[1] Canadian National Collection of Insects, Arachnids and Nematodes, Agriculture and Agri-Food Canada, Ottawa, ON, Canada
[2] Plant Protection Research Department, South Agricultural and Natural Resources Research and Education Center, AREEO, Birjand, Iran
[3] Carleton University, Biology Department, Ottawa, ON, Canada

## ABSTRACT

Species of the distinctive and cosmopolitan genus *Dasydorylas* Skevington, 2001 in the Middle East are revised. Seven species are documented, and three new species, *Dasydorylas dactylos* sp. nov., *D. forcipus* sp. nov. and *D. parazardouei* sp. nov., are described, and one synonym, *D. derafshani* Motamedinia & Kehlmaier, 2017, syn. nov. is proposed, based on sequence information from the mitochondrial COI barcoding gene and morphological parameters. Diagnoses, illustrations and distributional data are provided for all studied species. Descriptions of new species as well as an identification key to all known species in the Middle East are also provided.

## INTRODUCTION

*Dasydorylas* Skevington, 2001 are mid-sized (2.8–4.5 mm) big-headed flies placed within the tribe Eudorylini (Diptera: Pipunculidae: Pipunculinae). The first *Dasydorylas* species were described by *Becker (1897)* as *Pipunculus horridus* and *P. discoidalis*. *Perkins (1905)*, *Becker (1908)*, *Banks (1915)*, *Hardy (1950*, *1954*, *1961*, *1968*, *1972)*, *Koizumi (1959)* and *Kuznetzov (1994)* described additional *Dasydorylas* species under the genera *Pipunculus*, *Dorilas* and *Eudorylas*. *Dasydorylas* was coined during a comprehensive phylogenetic study of world Eudorylini published by *Skevington & Yeates (2001)*. Although *Skevington & Yeates (2001)* found that *Dasydorylas* is sister to a large clade including *Amazunculus Rafael, 1986*, *Elmohardyia Rafael, 1987b*, *Basileunculus Rafael, 1987a*, *Allomethus Hardy, 1943* and *Claraeola Aczél, 1940*, our preliminary unpublished target enrichment analysis suggests that the genus is sister to *Claraeola Aczél, 1940*. The genus group can be best distinguished by a combination of the following morphological characters: pterostigma present, notopleuron often with dense bush of long setae, femora often with posterdorsal row of long and back setae, scutellum with an apical setae, tergite 1 with distinct setae laterally, sternite 3–5 often with posterior setae, syntergosternite eight with membranous area of medium size, male terminalia with ejaculatory apodeme funnel-shaped (*Skevington & Yeates, 2001*). There are few biological

Corresponding author
Behnam Motamedinia,
bmoetamedi@yahoo.com

data available for the genus *Dasydorylas*. *Dasydorylas eucalypti* (*Perkins, 1905*) (type species) and *D. comitans* (*Perkins, 1905*) were reared from Cicadellidae nymphs (Hemiptera: Auchenorrhyncha). *Dasydorylas* is a cosmopolitan genus with 32 described species occurring in the Afrotropical (nine species), Palaearctic (nine species), Australasian and/or Oriental (nine species) and the Nearctic and/or Neotropical regions (five species) (*Skevington & Yeates, 2001*; *Kehlmaier, 2005a*, *2005b*; *Földvári, 2013*; *Motamedinia et al., 2017a*, *2017b*; J. Skevington, 2019, unpublished data). An identification key to Palaearctic and Afrotropical species was provided by *Kehlmaier (2005a)* and *Földvári (2013)*.

Despite its large size (7,207,575 km$^2$) and special geographical position, the Middle East (here defined to include Bahrain, Cyprus, Iran, Iraq, Israel, Jordan, Kuwait, Lebanon, Oman, Palestine, Qatar, Saudi Arabia, Syria, Turkey, United Arab Emirates and Yemen) *Dasydorylas* diversity is largely unknown and only three species (*D. discoidalis* (*Becker, 1897*); *D. gradus Kehlmaier, 2005b*; *D. zardouei Motamedinia et al., 2017a*) have been reported from this poorly studied region.

In this work we revise the genus *Dasydorylas* in the Middle East and describe three new species based on morphological characteristics and sequence data from the mitochondrial COI barcoding region. We also provide an identification key and distribution map to the Middle Eastern *Dasydorylas* species.

## MATERIALS AND METHODS

### Specimen collection and morphological study

The specimens examined were collected by Malaise trap and sweep net from Iran, Israel, the United Arab Emirates and Yemen. Specimens captured in traps were collected into 60–90% ethanol, dehydrated into absolute ethanol and then critical-point dried (CPD) and pointed. The *Dasydorylas* specimens examined for this study belong to the following collections: Canadian National Collection of Insects (CNC), Arachnids and Nematodes, Hayk Mirzayans Insect Museum (HMIM), Insect Taxonomy Research Department, Iranian Research Institute of Plant Protection, Tehran, Iran and Tel Aviv University (TAU), Israel.

For several species, characters for accurate identification have only been found in the male genitalia. We applied a morphological species concept for our work, essentially looking for discrete differences in the male genitalia that were invariant between specimens examined. Some external characters were found to support the species concepts, but concepts were mostly supported empirically by DNA barcode data (i.e., significant barcode gaps were found between species and within species variation was usually less than 2%). No cryptic species were discovered based on DNA that could not be recognized morphologically (i.e., all BOLD BIN's were supported by both morphology and DNA). Male genitalia were stored in microvials containing glycerin after clearing in hot lactic acid (85%) at 100 °C for 30–240 min. For some darker genitalia, terminalia were treated with 10% KOH at 100 °C for 10–30 min then neutralized in glacial acetic acid for 5 min. Females were included in the type series if DNA data corroborated their linkage with sequenced males. All specimens are labeled with a unique reference number from the CNC

database (e.g., Jeff_Skevington_Specimen12345 and CNC_Diptera12345, Abbreviated as JSS12345 and CNCD12345 respectively) and can be accessed at https://cnc.agr.gc.ca/. SimpleMappr (*Shorthouse, 2010*) was used to create the species distribution map.

Photographic equipment used to visualize external characters was a Leica DFC450 module fitted on a Leica M205C stereomicroscope and 0.6x lens. Image series comprising 15–20 focal planes were merged to produce a single image with increased depth of field using the image-stacking software ZereneStacker (*Littlefield, 2018*). Dissected material was mounted in glycerin and photographed with a Leica DM5500B microscope equipped with a Leica DMC4500 module connected to a personal computer running the Leica Application Suite software (https://www.leica-microsystems.com), which includes an Auto-Montage module that combines multiple layers of photographs into a single fully focused image. Photos were edited and finalized in Adobe Photoshop CS3® imaging software. The terminology used in the descriptions is based on *Skevington (2002)* and *Kehlmaier (2005a)* with the following abbreviations being used throughout the article: LF: WF, ratio of length of flagellum to its width; LW:MWW, ratio of length of wing to maximum width of wing; LS:LTC, ratio of length of pterostigma to length of third costal segment; LTC:LFC, ratio of length of third costal segment to length of fourth costal segment; LT35:WT5, ratio of length of tergites 3–5 to maximum width of tergite 5; WT5: LT5, ratio of width of tergite 5 to its length; T5R:T5L, ratio of length of right margin of tergite 5 to length of its left margin; LT35:WS8, ratio of length of tergites 3–5 to width of syntergosternite 8; LS8:HS8, ratio of length syntergosternite 8 to its height; MLE:MWE, ratio of maximum length of epandrium to its maximum width (viewed dorsally); LP:LB, ratio of length of piercer to length of base (viewed laterally); LDP:LPP, ratio of length of distal part of piercer to length of its proximal part (viewed laterally).

## DNA extraction, PCR amplification and sequencing

Total genomic DNA was nondestructively extracted from two legs, or whole specimens using the DNeasy Blood & Tissue kit (Qiagen Inc., Santa Clara, CA, USA) following the manufacturer's protocol. Following extraction, specimens were CPD'ed and deposited as vouchers in one of the aforementioned collections.

For DNA barcoding, a 658 bp fragment of the 5′ end of the mitochondrial coding gene cytochrome oxidase subunit I (COI) was sequenced following protocols published by *Gibson et al. (2011)*. In some cases, initial attempts to amplify the full COI barcode failed, presumably due to the degradation of the DNA. In these cases, a novel COI mini-barcode protocol was employed (A.D. Young, 2020, in preparation) in order to amplify a 214 bp fragment (COI-Fx-C), located at the 3′-end of the COI barcode region, for species identification. In the case of putative new species, efforts were made to amplify the 5′ and middle COI mini-barcode fragments (COI-Fx-A and COI-Fx-B respectively) that, when combined, provide a complete COI barcode sequence. Oligonucleotides (primers) used in this study are listed in Table 1. Sanger Sequencing was performed at CNC.

All sequence chromatograms were edited and contigs formed using Sequencher 5.4.6 (Gene Codes Corp., Ann Arbor, MI, USA). Resulting contigs were hand-aligned using Mesquite 3.6 (*Maddison & Maddison, 2018*). Uncorrected pairwise genetic distances

**Table 1 Cytochrome c Oxidase I mitochondrial gene primers.**

| Gene name/region | Forward primer name | Forward primer sequence (5′-3′) | Primer reference | Reverse primer name | Reverse primer sequence (5′-3′) | Primer reference |
|---|---|---|---|---|---|---|
| COI Barcode | LCO1490 | GGTCAACA AATCATAAA GATATTGG | *Folmer et al. (1994)* | COI-Dipt-2183R | CCAAAAAATC ARAATARRTG YTG | *Gibson et al. (2011)* |
| COI-Fx-A (5′ end of barcode) | LCO1490 | GGTCAACA AATCATAAA GATATTGG | *Folmer et al. (1994)* | COI-SYR-1762R | CGDGGRAAD GCYATRTCDGG | A.D. Young, 2020 (in preparation) |
| COI-Fx-B (middle of barcode) | COI-SYR-342F | GGDKCHCC NGAYATRGC | A.D. Young, 2020 (in preparation) | COI-SYR-1976R | GWAATRAART TWACDGCHCC | A.D. Young, 2020 (in preparation) |
| COI-Fx-C (3′ end of barcode) | COI-SYR-1957F | GGDATWTC HTCHATYYTAGG | A.D. Young, 2020 (in preparation) | COI-Dipt-2183R | CCAAAAAATCA RAATARRTGYTG | *Gibson et al. (2011)* |

(p-distance) were calculated with Mega7 (*Kumar, Stecher & Tamura, 2016*). Sequence Accession Numbers issued by GenBank (GB) are provided for each species in the material examined sections.

The electronic version of this article in Portable Document Format will represent a published work according to the International Commission on Zoological Nomenclature (ICZN), and hence the new names contained in the electronic version are effectively published under that Code from the electronic edition alone. This published work and the nomenclatural acts it contains have been registered in ZooBank, the online registration system for the ICZN. The ZooBank Life Science Identifiers (LSIDs) can be resolved and the associated information viewed through any standard web browser by appending the LSID to the prefix http://zoobank.org/. The LSID for this publication is: urn:lsid: zoobank.org:pub:A19B5B2E-817F-463C-A217-869C44C25C0A. The online version of this work is archived and available from the following digital repositories: PeerJ, PubMed Central and CLOCKSS.

# RESULTS

**Taxonomy**

Genus *Dasydorylas* *Skevington & Yeates, 2001*

Type species: *Pipunculus eucalypti* *Perkins, 1905*

**Diagnosis**

Small to medium bodied flies; body length 2.8–4.5 mm; wing length 2.7–5.2 mm; pedicel with 3–6 upper and 1–4 lower bristles; flagellum silver gray to brownish pollinose; vertex lacking pollinosity bearing an elevated equilateral ocellar triangle; postpronotal lobe often with 4–7 long setae along upper margin; scutellum with a fringe of 6–16 setae; front femur usually with rows of ventral spines; front and mid tibiae with short distal spines; pterostigma present; crossvein r-m reaches dm at or after one third of the cells length; abdomen ovate, ground color dark (in some specimens with posterolateral markings of gray pollinosity); tergite 1 with 3–16 long bristles laterally; hypandrium often with cluster of anteromedial setae; phallus trifid; phallic guide strong (some species with medium to

large spines); ejaculatory apodeme usually funnel-shaped; sperm pump usually vase-shaped, with lateral flange around entire upper surface.

**Biology**
Unknown except *D. eucalypti* and *D. comitans*, which were reared from Cicadellidae nymphs (*Perkins, 1905*).

**Distribution**
Afrotropical (Botswana, Burundi, Congo (Democratic Republic), Kenya, Madagascar, Malawi, Morocco, Namibia, Tanzania, Uganda, United Arab Emirates, Zimbabwe); Australasia (Australia, Papua New Guinea); Nearctic (Mexico, USA); Neotropical (Argentina, Brazil, Costa Rica, Nicaragua); Oriental (India, Laos, Myanmar, Philippines, Taiwan, Thailand, Vietnam); Palaearctic (Austria, Belgium, Bulgaria, Canary Islands, China, Croatia, Czech Republic, Denmark, France, Germany, Great Britain, Hungary, Iran, Israel, Italy, Japan, Latvia, Netherlands, North Korea, Portugal, Romania, Russia, Slovakia, Spain, Switzerland) (*Kehlmaier, 2005a*, *2005b*; *Kehlmaier, Gibbs & Withers, 2019*; *Motamedinia et al., 2017a*, *2017b*; J. Skevington, 2019, unpublished data).

## Key to males of *Dasydorylas* species in the Middle East

1 Compound eyes meeting each other (Figs. 1A and 1B) .............................. 3

—Compound eyes converging but not meeting each other (Figs. 2A and 2B) .......... 2

2 Phallic guide with 8 downward spines (Fig. 3C); base of surstyli longer than wide (Figs. 3A and 3B) .............................................. ***D. discoidalis* (Becker)**

—Phallic guide with 6–7 downward spines, base of surstyli as long as wide (Figs. 4A and 4B) ...................................... ***D. dactylos* Motamedinia & Skevington**

3 Abdominal tergite 1 with 13–16 strong bristles laterally (Figs. 5A and 5B); phallic guide without spine (Figs. 6D and 6E) .................................. ***D. horridus* (Becker)**

—Abdominal tergite 1 with fewer than 13 bristles laterally; phallic guide with spine (Figs. 7D and 8D) ......................................................................... 4

4 Phallic guide with 13–14 long spines (Figs. 8D and 8E).......... ***D. gradus* Kehlmaier**

—Phallic guide with two spines ...................................................... 5

5 Hind femur with weak wrinkles anteriorly (Fig. 9A); inner side of surstyli almost rounded in lateral view (Figs. 7C and 7D)...... ***D. forcipus* Motamedinia & Skevington**

—Hind femur without anterior wrinkles, inner side of surstyli not as above in lateral view............................................................................ 6

6 Abdomen brown (Fig. 1A); hypandrium with hypandrial apodeme (Fig. 10B) ....... ........................................ ***D. parazardouei* Motamedinia & Skevington**

—Abdomen dark, hypandrium without hypandrial apodeme (Fig. 11B) ............ .................................................. ***D. zardouei* Motamedinia & Kehlmaier**

***Dasydorylas dactylos* Motamedinia & Skevington sp. nov.**
Figures 4A–4E
urn:lsid:zoobank.org:act:BCE6B5FC-5C25-49C4-8473-F6D662DCE8CF

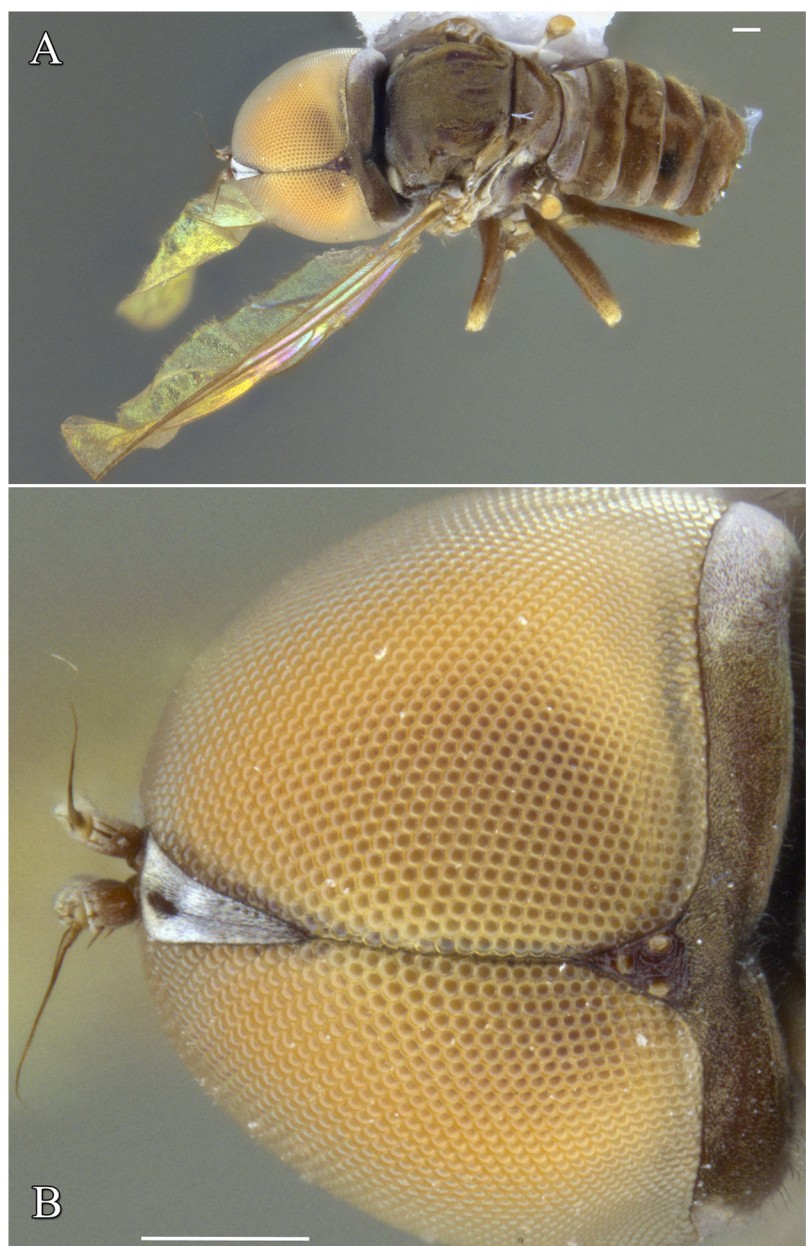

**Figure 1 Male of *Dasydorylas parazardouei* Motamedinia & Skevington sp. nov. (A) habitus in dorsal view, (B) compound eyes in dorsal view.** Scale bar = 0.25 mm. Images by the authors.

## Materials examined

Israel: holotype: male, Neot Semadar, 30.0333, 35.0166, 4.XII.1995, leg. A. Freidberg, JSS50777, GB: MN520769, TAU.

## Diagnosis

This species can be recognized by the long tapering flagellum; separated compound eyes in males; ocellar triangle divided by a median groove; apical finger-like process in surstyli; 6–7 downward spines on either side of phallic guide (Figs. 4D and 4E).

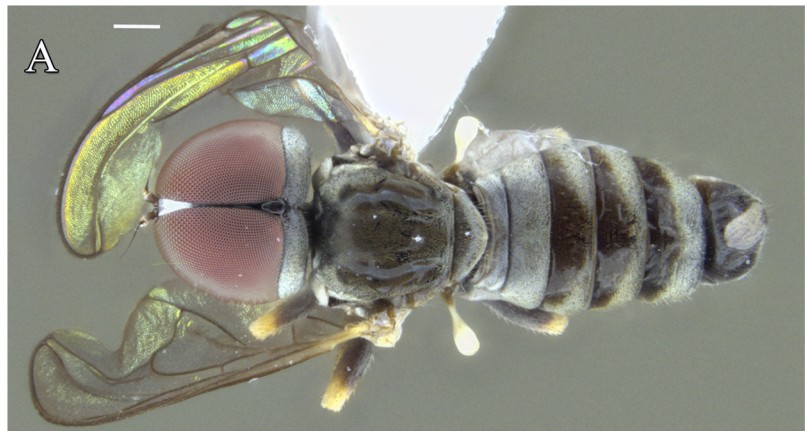

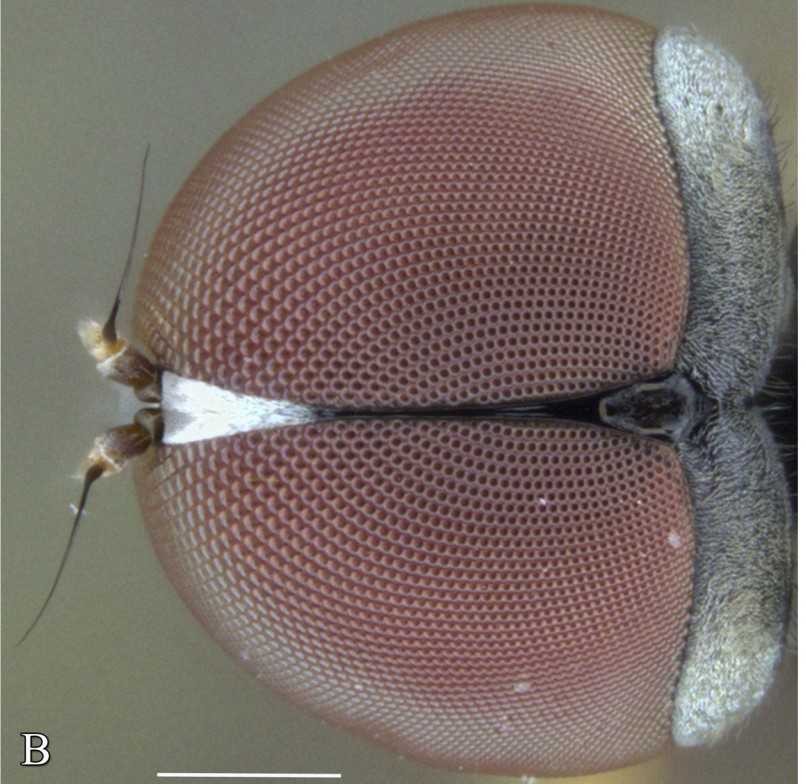

**Figure 2  Male of *Dasydorylas discoidalis* (A) habitus in dorsal view, (B) compound eyes in dorsal view.** Scale bar = 0.25 mm. Images by the authors.

## Description

Body length. 3.3 mm (excluding antennae).

**Head.** Face dark, silver-gray pollinose. Scape, pedicel and arista dark; pedicel with 1–2 short upper bristles and two short lower bristles; flagellum light brown, long tapering (LF:WF = 2.4). Eyes converging but not meeting and separated by less than diameter of frontal facets. Frons dark, silver-gray pollinose; vertex dark with elevated equilateral ocellar triangle divided by a median groove, lacking pollinosity, shining black; occiput dark,

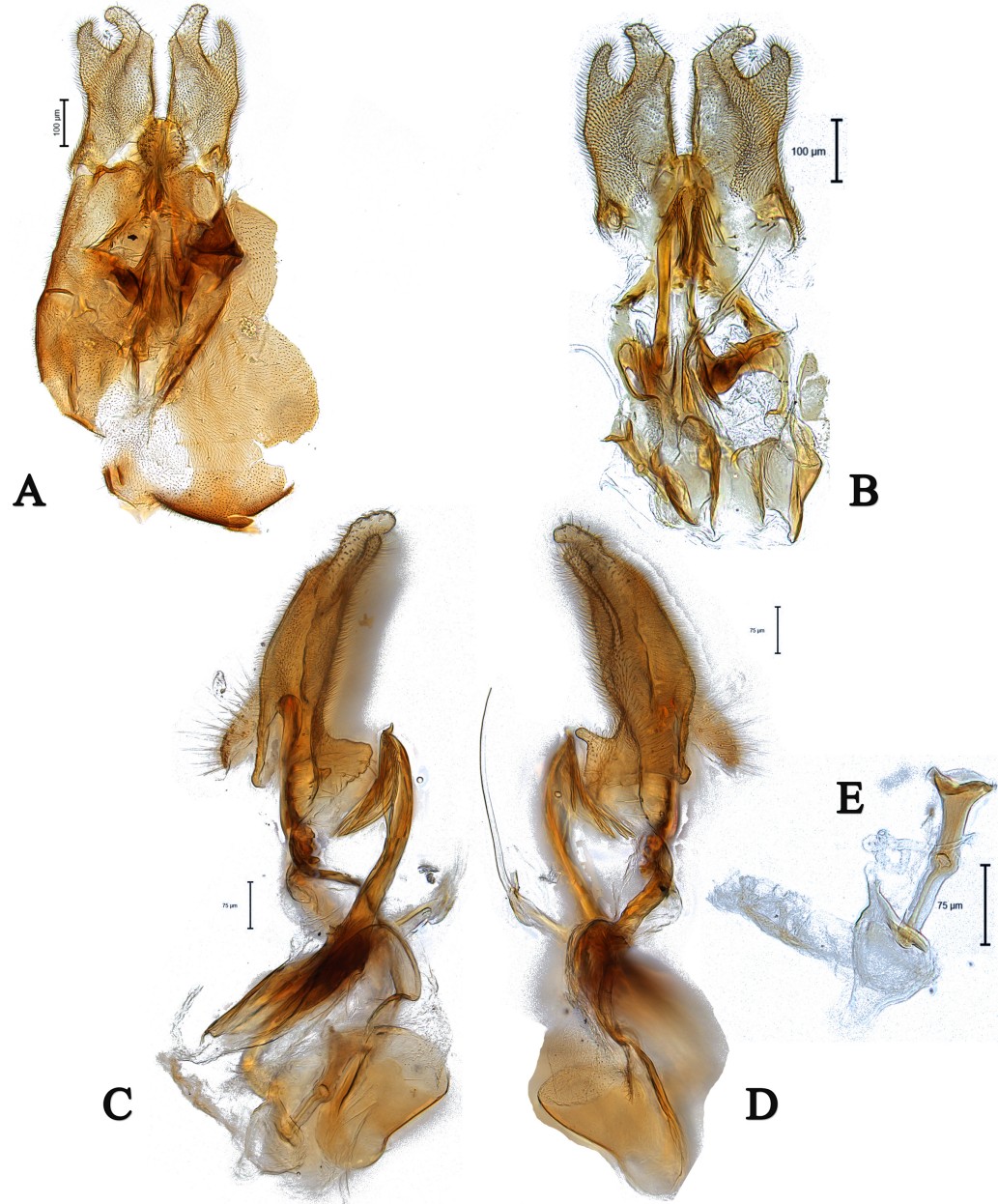

**Figure 3 Male genitalia of *Dasydorylas discoidalis* (A) in dorsal view, (B) in ventral view, (C and D) in lateral view, (E) ejaculatory apodeme.** Images by the authors.

gray pollinose. **Thorax.** Pleura, prescutum, scutum and scutellum dark. Pleura gray pollinose. Postpronotal lobe pale, gray pollinose and with 10–12 postpronotal setae along upper margin. Prescutum and scutum gray pollinose, with two uniseriate dorsocentral rows of setae and some supra-alar setae. Scutellum gray-brown pollinose, with a fringe of up to 16 long dark hairs (up to 0.18 mm) and with numerous shorter hairs on its dorsal surface. Subscutellum gray-brown pollinose. **Wing.** Length: 3.5 mm.

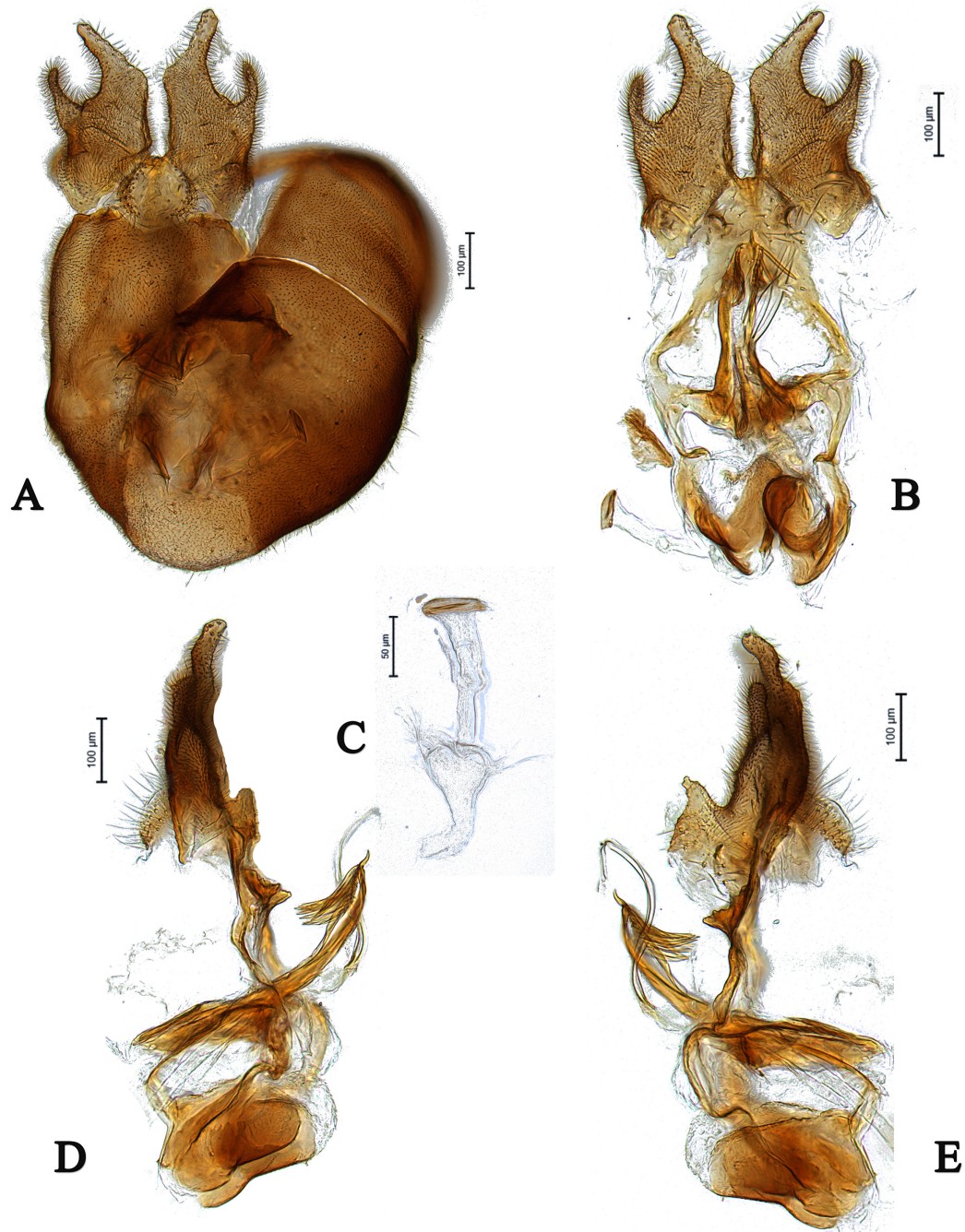

**Figure 4 Male genitalia of *Dasydorylas dactylos* Motamedinia & Skevington sp. nov. (A) in dorsal view, (B) in ventral view, (C) ejaculatory apodeme, (D and E) in lateral view.** Images by the authors.

LW:MWW = 3.3. Wing almost entirely covered with microtrichia. Pterostigma brown and complete (LS:LTC = 1.0, LTC:LFC = 1.1). $M_1$ straight. Length of halter: 0.5 mm; base dark, stem narrowly white and knob yellow; base and stem somewhat gray pollinose. **Legs.** Fore and mid coxae dark brown, hind coxa dark, gray pollinose; mid coxa with 4–5 brown

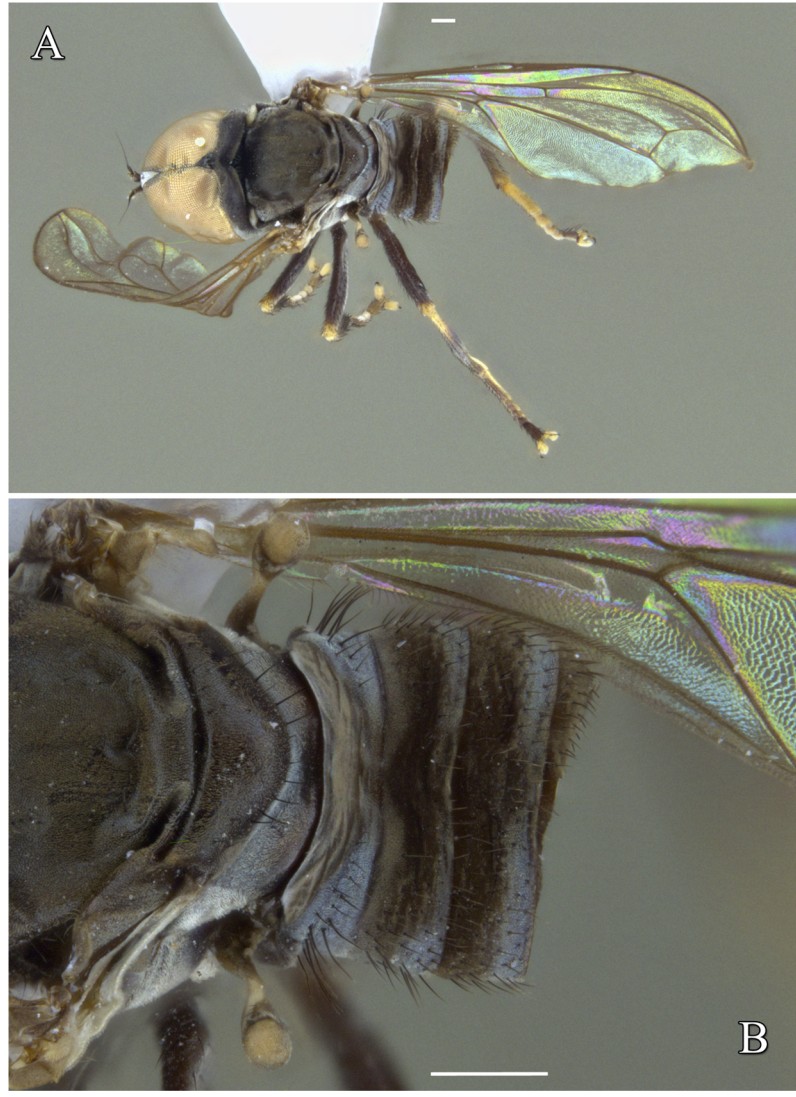

**Figure 5** Male of *Dasydorylas horridus* (A) habitus in dorsal view, (B) thorax and abdomen in dorsal view. Scale bar = 0.25 mm. Images by the authors.

anterior bristles. Trochanters dark brown, gray pollinose; hind trochanter with 1–2 long lateral bristles. Femora dark brown, distinctly yellow at apex, gray pollinose. All femora bearing two rows of dark, smaller, peg-like anteroventral spines on apical one third; hind femur swollen in middle. Tibiae yellow, ventrally darkened in apical half, with three rows of yellow setae on anterior and posterior side, without apical spines. Tarsi yellow, brown pollinose. Distitarsi dark. Pulvilli longer than distitarsi. **Abdomen.** Ground color dark. Tergite 1 with 8–10 light brown lateral bristles. Tergites 1–5 gray pollinose with scattered light brown bristles. Syntergosternite 8 dark brown, brown pollinose, without dorsal depression on side of right surstylus. Membranous area vertically directed, broader in upper half, occupying about a third of the width of syntergosternite 8. **Genitalia.** Genital

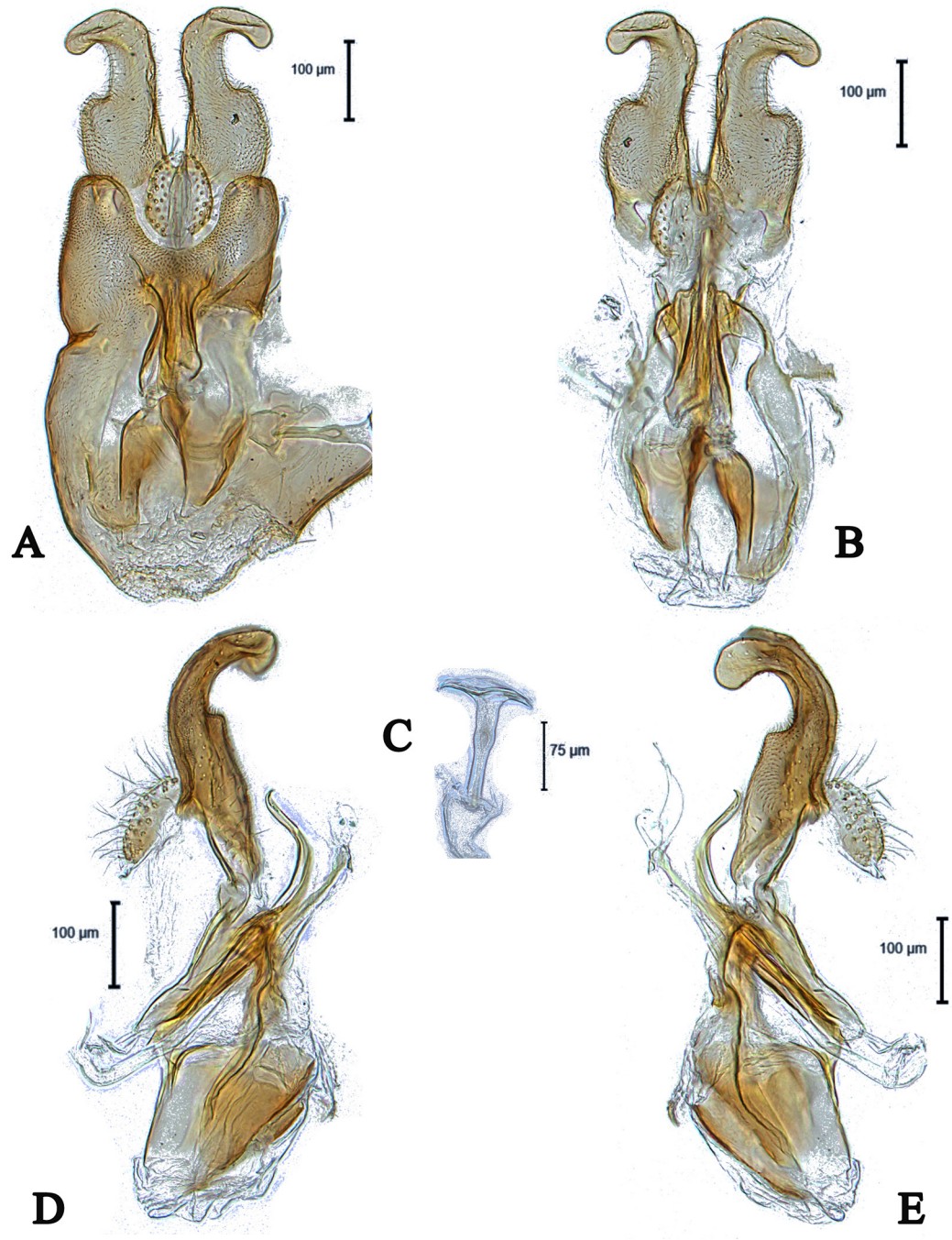

**Figure 6 Male genitalia of *Dasydorylas horridus* (A) in dorsal view, (B) in ventral view, (C) ejaculatory apodeme, (D and E) in lateral view.** Images by the authors.

capsule in dorsal view: epandrium dark brown, brown pollinose and longer than wide (MLE:MWE = 1.1). Surstyli brown, brown pollinose and symmetrical. Both surstyli have a broad and short base with an inner, long apical fingerlike process (Fig. 4A). Tips of outer

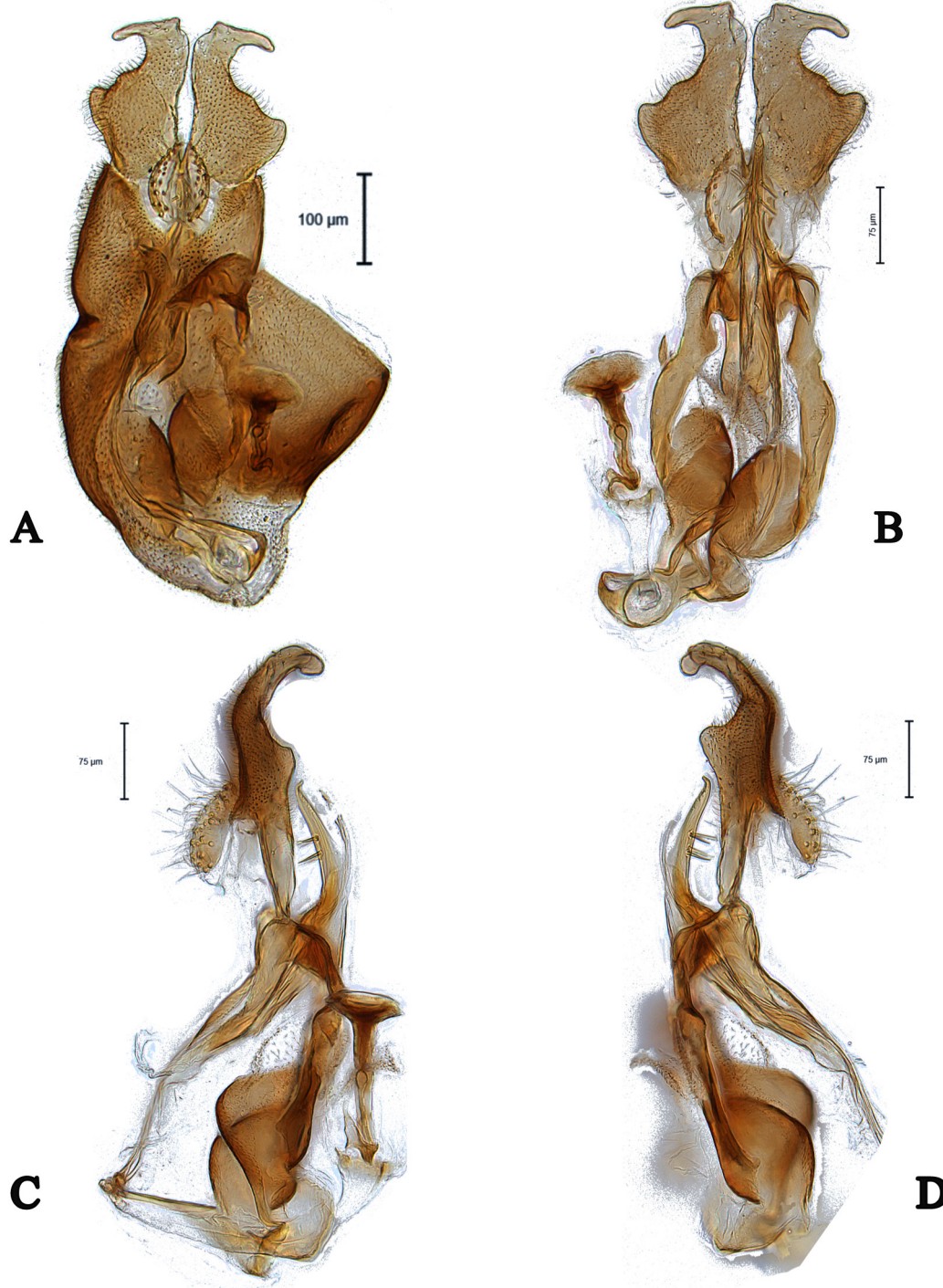

**Figure 7 Male genitalia of *Dasydorylas forcipus* Motamedinia & Skevington sp. nov. (A) in dorsal view, (B) in ventral view, (C and D) in lateral view.**

projection slightly bent inward. Genital capsule in ventral view: gonopods minute and symmetrical (Fig. 4B). Genital capsule in lateral view: both surstyli in basal half broad, in apical half narrowed to form a finger-like process, outer one is straight and inner one bent

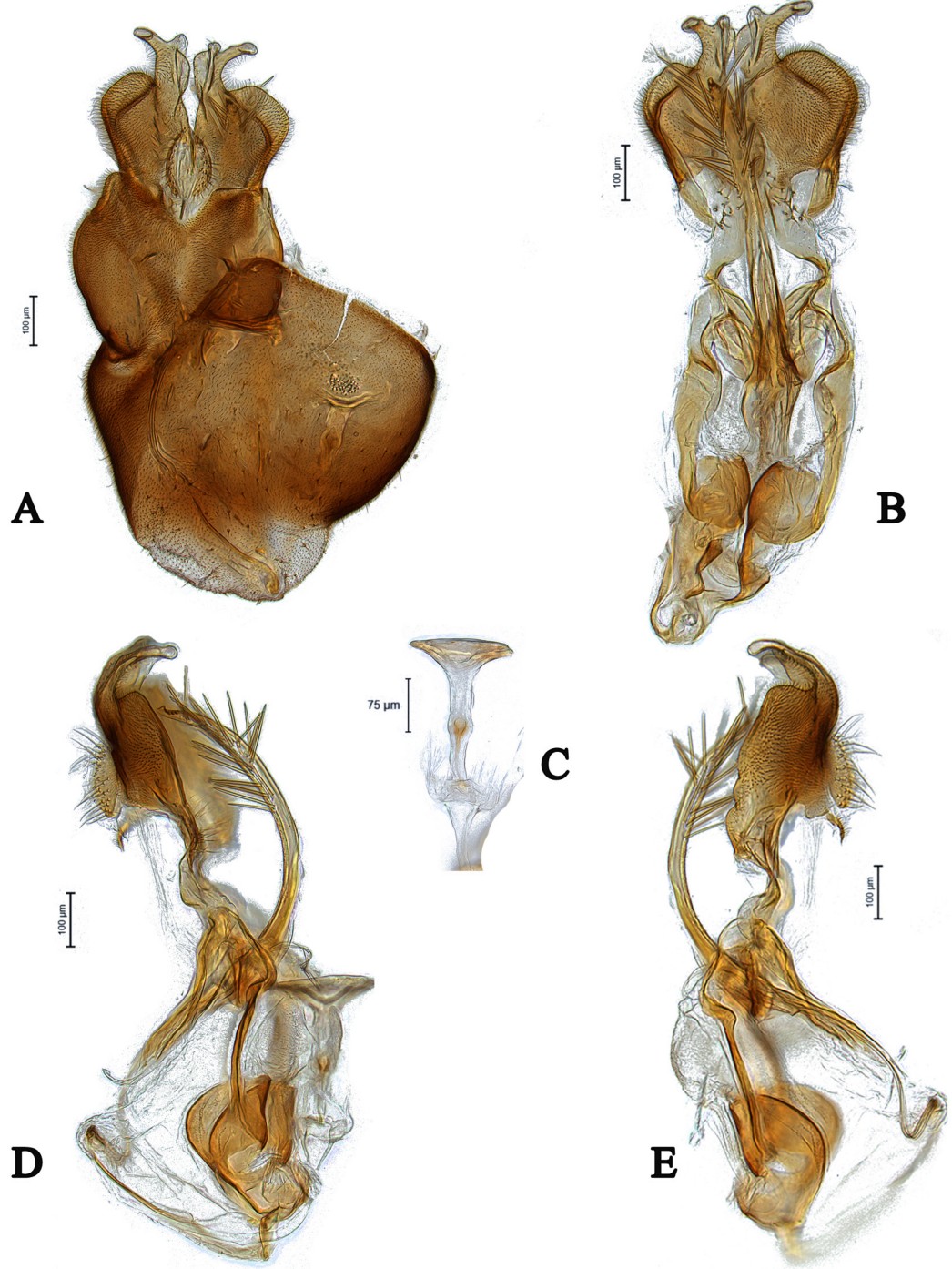

**Figure 8 Male genitalia of *Dasydorylas gradus* (A) in dorsal view, (B) in ventral view, (C) ejaculatory apodeme, (D and E) in lateral view.**

slightly towards the sternite (Figs. 4D and 4E). Phallus trifid, straight and long, phallic guide of medium length, broad with fingerlike process at apex, bow-like bent towards surstyli, on either side with 6–7 downwards directed long spines at its apex (Figs. 4D and 4E). Ejaculatory apodeme funnel-shaped (Fig. 4C).

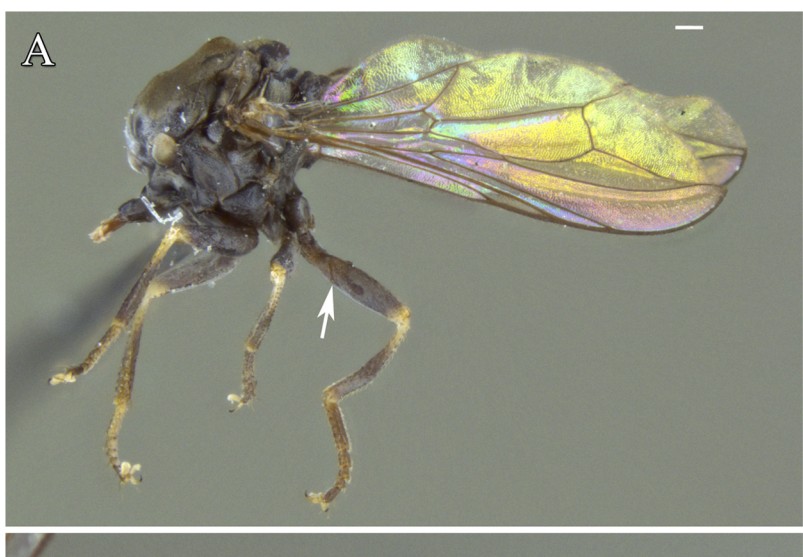

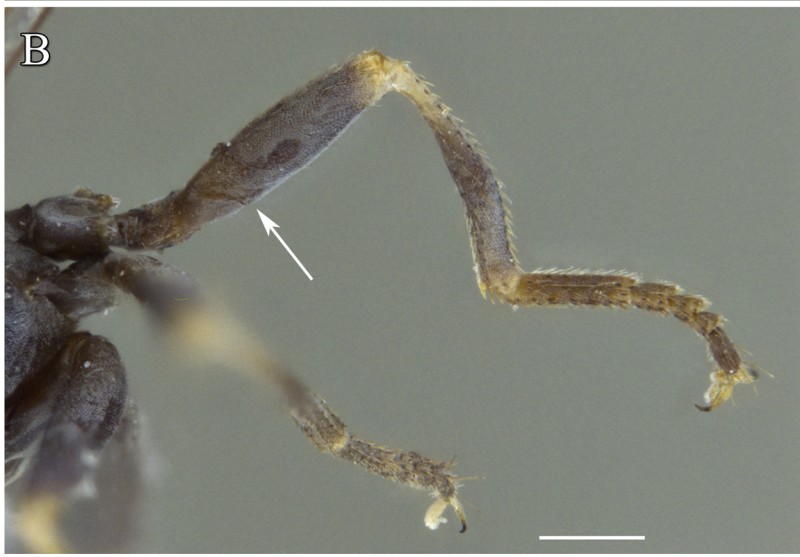

**Figure 9 Male of *Dasydorylas forcipus* Motamedinia and Skevington sp. nov. (A) habitus in lateral view, (B) hind leg in anterior view.** Scale bar = 0.25 mm.

**Distribution**
Israel (Fig. 12).

**Etymology**
The species name is derived from Greek "dactylos" (finger) referring to the shape of the surstyli.

**Molecular variation**
Based on uncorrected pairwise genetic distances (p-distance), this species is close to *D. discoidalis*, differing by 6.9% (Table 2).

***Dasydorylas discoidalis*** (*Becker, 1897*)
Figures 2A, 2B and 3A–3E

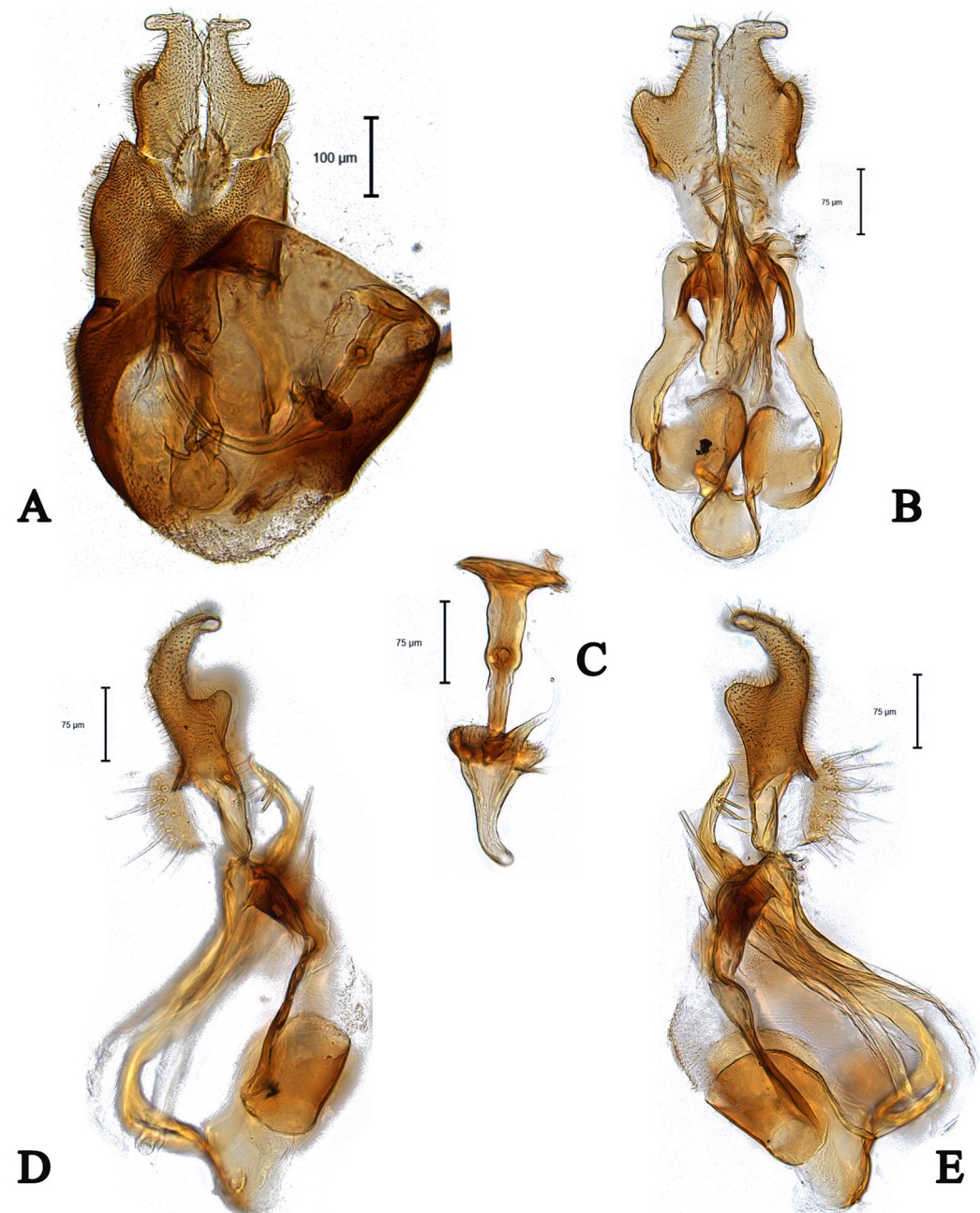

**Figure 10 Male genitalia of *Dasydorylas parazardouei* Motamedinia and Skevington sp. nov. (A) in dorsal view, (B) in ventral view, (C) ejaculatory apodeme, (D and E) in lateral view.**

*Pipunculus discoidalis* Becker, 1897: 46.
*Dasydorylas derafshani* Motamedinia et al., 2017a; **syn. nov.**

**Materials examined**
United Arab Emirates: two males, Abu Dhabi, Al Wathba Wetland Reserve, 24.254303, 54.610875, II.2015, leg. A. Saji & A. van Harten, Malaise trap, CNCDD470470,

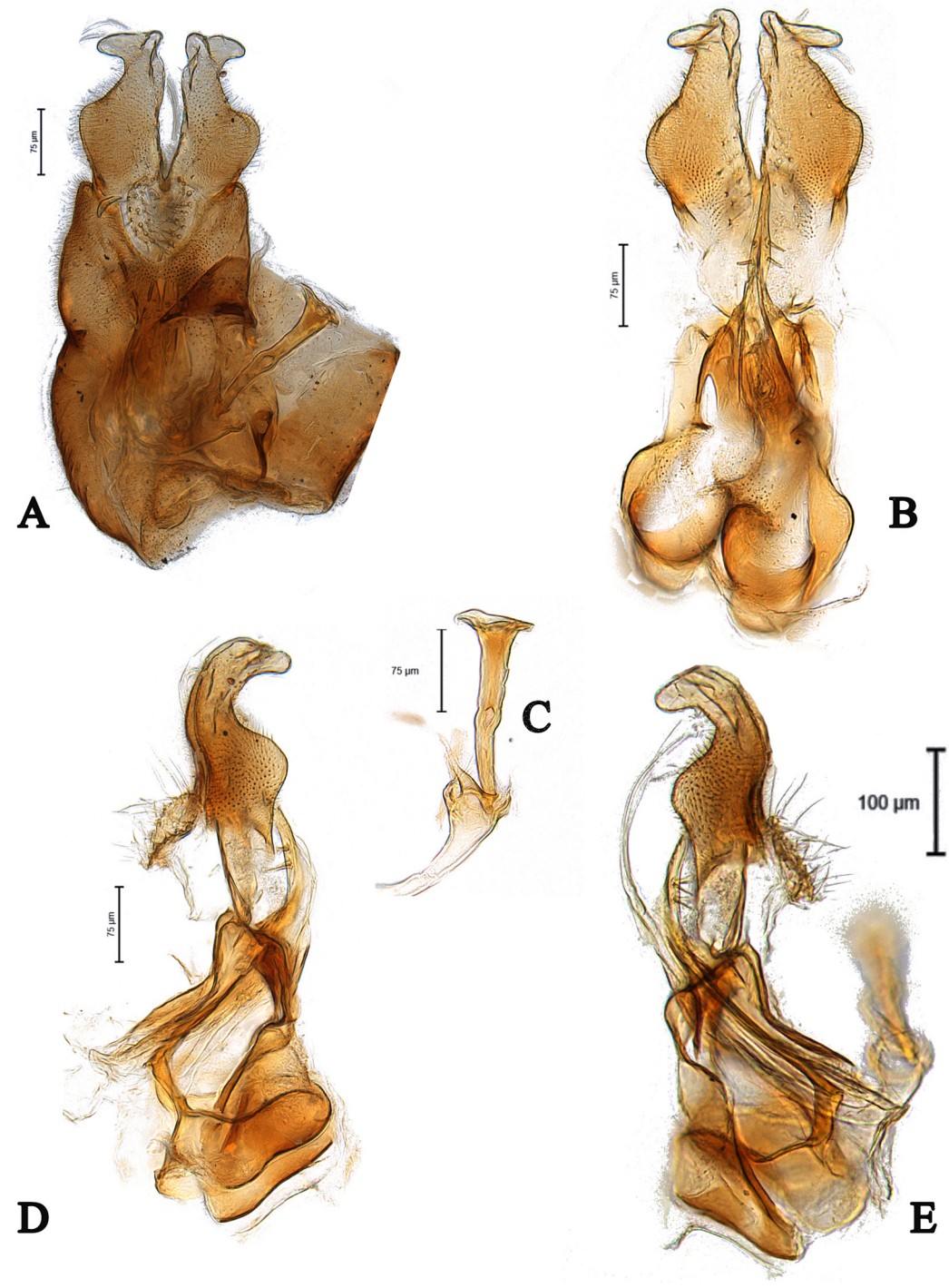

**Figure 11 Male genitalia of *Dasydorylas zardouei* (A) in dorsal view, (B) in ventral view, (C) ejaculatory apodeme, (D and E) in lateral view.**

CNCD470534, CNCD; six males, four females, III.2015, leg. A. Saji & A. van Harten, Malaise trap, CNCD470632, GB: MN520770, CNCD470633, CNCD470635, CNCD470647, GB: MN520764, CNCD470655, CNCD470658, CNCD470659,

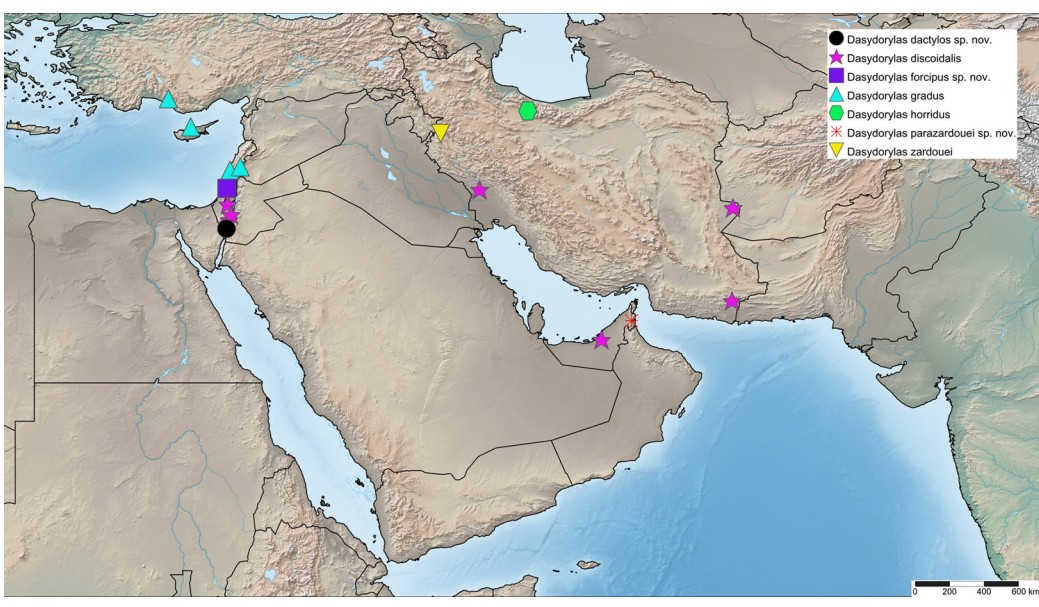

**Figure 12** *Dasydorylas* distribution in the Middle East.

**Table 2 Uncorrected pairwise distances among *Dasydorylas* species in the Middle East (intraspecific distances are highlighted in bold).**

| | | 1 | 2 | 3 | 4 | 5 | 6 | 7 | 8 | 9 | 10 | 11 |
|---|---|---|---|---|---|---|---|---|---|---|---|---|
| 1 | *D. horridus*-JSS52204 | | | | | | | | | | | |
| 2 | *D. dactylos*- JSS50777*(AC) | 0.134 | | | | | | | | | | |
| 3 | *D. discoidalis*-JSS52200 | 0.151 | 0.099 | | | | | | | | | |
| 4 | *D. discoidalis*-CNCD470632 | 0.142 | 0.086 | **0.013** | | | | | | | | |
| 5 | *D. discoidalis*-CNCD470809 | 0.142 | 0.086 | **0.013** | **0.000** | | | | | | | |
| 6 | *D. gradus*-JSS50850 | 0.099 | 0.095 | 0.134 | 0.121 | 0.121 | | | | | | |
| 7 | *D. gradus*-JSS52303 | 0.103 | 0.091 | 0.129 | 0.116 | 0.116 | **0.004** | | | | | |
| 8 | *D. gradus*-JSS50771*(AC) | 0.108 | 0.082 | 0.147 | 0.134 | 0.134 | **0.013** | **0.017** | | | | |
| 9 | *D. parazardouei*-CNCD175301*(C) | 0.142 | 0.147 | 0.168 | 0.168 | 0.168 | 0.112 | 0.108 | 0.099 | | | |
| 10 | *D. parazardouei*-CNCD175303*(C) | 0.142 | 0.147 | 0.168 | 0.168 | 0.168 | 0.112 | 0.108 | 0.099 | **0.000** | | |
| 11 | *D. discoidalis*-CNCD470647*(C) | 0.155 | 0.069 | **0.030** | **0.017** | **0.017** | 0.129 | 0.125 | 0.116 | 0.151 | 0.151 | |

**Note:**
\* Specimen sequence data was obtained using the COI mini-barcode protocol. A and C denote the COI mini-barcode regions sequenced.

CNCD470661, CNCD470696, CNCD470700, CNCD; four males, seven females, IV.2015, leg. A. Saji & A. van Harten, Malaise trap, CNCD470809, GB: MN520762, CNCD470813, CNCD470814, CNCD470815, CNCD470818, CNCD470820, CNCD470822, CNCD470826, CNCD470836, CNCD470843, CNCD470848, CNC. Israel: one male, Arava Valley, nr Hazeva, Shizaf Nature Reserve, side channel of Wadi Shahak, 30.7500, 35.2500, −116 m, 24.III.1995, leg. M. E. Irwin, Malaise trap, JSS50852, TAU; one male, Enot Zuqim south to gate, 30.4833, 35.1500, 26.IV.2006, leg. L. Friedman, JSS50829, TAU; one male, Nahal Deragot, 31.3000, 35.0833, 790m, 28.II.1994, leg. A.

Freidberg, JSS50851, TAU. Iran: 1♂, Khuzestan, Shush, 32.066667, 48.2333, 68m, 11.
III–10.V.2015, leg. E. Gilasian, Malaise trap, JSS52200, GB: MN520767, HMIM; one
female, Sistan & Balochestan, Rask, 26.266667, 61.416667, 139m, 10.VI–14.VII.2016, leg.
M. Ghaforimoghadam, Malaise trap, JSS52140, CNC; one male, Zabol, 31.116667,
61.466667, 481m, 15.VI.2016, leg. H. Derafshan, sweeping, JSS51901, CNC; one female, 6.
VI.2016, leg. H. Derafshan, sweeping, JSS51873, CNC.

## Diagnosis

This species can be recognized by separated compound eyes in males (converging but not
meeting) (Figs. 2A and 2B); long posterior setal fringe of the scutellum and evenly
distributed setae on the abdominal tergites; phallic guide with eight downward directed
spines at the apex (Figs. 3C and 3D); ovipositor with the largely swollen sternite 8.

## Distribution

Iran, Israel, Russia, United Arab Emirates (*Kehlmaier, 2005a*; *Motamedinia et al., 2017b*;
J. Skevington, 2019, unpublished data) (Fig. 12 shows Middle Eastern distribution only).

## Notes

*Dasydorylas discoidalis* was described by *Becker (1897)* based on a female specimen.
Kehlamier re-described the female in *2005a*.

## Molecular variation

Sequence data for *D. discoidalis* (one female) and *D. derafshani* (three males) show they are
conspecific (0.0–3.0% uncorrected pairwise intraspecific difference—see Table 2). Based
on uncorrected pairwise genetic distances (p-distance), the nearest species to *D. discoidalis*
is *D. dactylus* with a COI distance of 6.9% (Table 2).

## Nomenclatural changes

*Dasydorylas derafshani Motamedinia et al., 2017a* is hereby treated as a new synonym of
*D. discoidalis Becker (1897)* based on the molecular evidence presented above.

### *Dasydorylas forcipus* Motamedinia & Skevington sp. nov.

Figures 7A–7D, 9A and 9B
urn:lsid:zoobank.org:act:0F5D0097-4983-41FB-B35E-E71AB00C56C0

## Materials examined

Israel: holotype : male, Nahal Qana Reserve, 32.1333, 35.0333, 120m, 9.VII.2007, leg.
A. Freidberg, JSS51680, TAU.

## Diagnosis

Hind femur with some weak wrinkles anteriorly (Figs. 9A and 9B); abdomen dark brown;
phallus trifid, shorter than phallic guide; phallic guide with two spines (Figs. 7B and 7D).

## Description

Body length. 2.9–3.0 mm (excluding antennae).

**Head.** Face dark, silver-gray pollinose. Scape dark, pedicel brown with two short upper bristles; flagellum dark brown, short tapering and gray pollinose (LF:WF = 1.7); arista dark with thickened base. Eyes meeting for 5–6 facets. Frons dark, silver-gray pollinose; vertex dark, lacking pollinosity, shining black; occiput dark, gray pollinose. **Thorax.** Pleura, prescutum, scutum and scutellum dark with a mixture of gray and brown pollinosity. Postpronotal lobe pale, gray pollinose and with 4–6 short postpronotal setae along upper margin. Prescutum and scutum with two uniseriate dorsocentral rows of setae and patches of supra-alar setae. Scutellum gray to brown pollinose, with a fringe of up to 12 short brown setae (up to 0.08 mm). Subscutellum with a mixture of gray and brown pollinosity. **Wing.** Length: 3.0–3.1 mm. LW:MWW = 2.7–2.8. Wing almost entirely covered with microtrichia. Pterostigma brown and complete (LS:LTC = 1.0, LTC:LFC = 1.18). $M_1$ straight. Length of halter: 0.4 mm; base dark, half of stem pale and knob brown; base and stem somewhat gray pollinose. **Legs.** Coxae dark, gray pollinose. Mid coxa and mid trochanter with two dark anterior bristles. Trochanters dark, partly gray pollinose. Femora dark, gray pollinose. Fore and mid femora bearing two rows of dark, small, peg-like anteroventral spines on apical one third. Hind femur with some weak wrinkles anteriorly. Tibiae dark, sometimes apices pale, gray pollinose, with three rows of setae on anterior and posterior side. Hind tibia with some weak wrinkles midanteriorly. Tarsi dark, gray pollinose, with some brown setae dorsally. Hind basitarsus as long as other tarsomeres. Distitarsi dark, longer than pulvilli. **Abdomen.** Ground color dark brown. Tergite 1 with five to six lateral bristles. Tergites with a mixture of gray and brown pollinosity. Syntergosternite 8 dark brown, brown pollinose. Membranous area large, roughly triangular, ventrocaudally directed. **Genitalia.** Genital capsule in dorsal view: epandrium dark brown, brown pollinose and wider than long (MLE:MWE = 0.8). Surstyli brown, pale at apices, brown pollinose, more reduced in apices, rather symmetrical. Both surstyli with a blocky base and a broad finger-like projection at its apical inner corner, bent outward distally by 90° (Fig. 7A). Genital capsule in ventral view: gonopods minute and symmetrical, with elongated regions of distinctly stronger sclerotization (Fig. 7B). Genital capsule in lateral view: epandrium without projecting lobe on either side. Both surstyli in basal half broad, in apical half narrowed to form a finger-like process, which is bent towards the sternite, inner side of surstyli almost rounded (Figs. 7C and 7D). Phallus trifid, straight and short; phallic guide bow-like bent towards surstyli, with two dorsolateral spines at the end of basal half on either side (Figs. 7C and 7D). Ejaculatory apodeme funnel-shaped (Fig. 7B).

### Distribution
Israel (Fig. 12).

### Etymology
The species name is derived from Latin forceps (tongs) referring to the shape of surstyli in the male genitalia.

### *Dasydorylas gradus* Kehlmaier, 2005b
Figures 8A–8E

**Materials examined**

Israel: one male, Nahal Namir, 33.0833, 35.2000, 2.XI.1998, leg. S. Alfi, JSS50771, GB: MN520768, TAU; one female, NabiHazuri, 33.2500, 35.7333, 790m, 18.X.2009, leg. A. Freidberg, JSS50850, GB: MN520763, TAU; Cyprus: one male, Kyrenia, 35.3477, 33.1504, 1–8.X.2017, leg. O. Ozden, Malaise trap, JSS52303, GB: MN520765, CNC; one male, 5–12. XI.2017, leg. O. Ozden, Malaise trap, JSS52307, CNC.

**Diagnosis**

Hind tibia with a wrinkled indentation midanteriorly, bearing one strong, dark bristle; abdominal tergite 1 with up to seven dark lateral bristles, becoming shorter towards the center of the tergite; phallic guide bow-like bent towards dorsal surface of genital capsule (in lateral view), with 13–14 long, straight spines, pointing upwards into various directions (Figs. 8D and 8E).

**Distribution**

Cyprus, Israel, Turkey, (*Kehlmaier, 2005b*; *Kehlmaier, Gibbs & Withers, 2019*; J. Skevington, 2019, unpublished data) (Fig. 12).

**Molecular variation**

Based on uncorrected pairwise genetic distances (p-distance), this species is close to *D. dactylus* differing by 8.2% (Table 2). Intraspecific genetic distance within the Israeli specimens is 1.3% and within Israeli and Cyprus specimens ranges from 0.4% to 1.7% (Table 2).

### *Dasydorylas horridus* (*Becker, 1897*)
Figures 5A, 5B and 6A–6E
*Pipunculus horridus Becker, 1897*: 41.

**Materials examined**

Iran: one male, Taleghan, Alborz, 36.166667, 50.7500, leg. A. Jabari, Malaise trap, JSS52204, GB: MN520766, CNC.

**Diagnosis**

Abdominal tergites densely covered with rather long setae; tergite 1 with about 13–16 strong and dark lateral bristles of different length; surstyli sickle shaped; phallic guide strictly bow-like bent towards surstyli, without spines (Figs. 6D and 6E); phallus trifid; ejaculatory apodeme funnel-shaped (Fig. 6C).

**Distribution**

Austria, Belgium, Bulgaria, Croatia, Czech Republic, France, Germany, Great Britain, Hungary, Iran, Italy, Latvia, Netherlands, Poland, Slovakia, Slovenia, Spain, Switzerland and Yugoslavia. (*Kehlmaier, 2005a*; *Kehlmaier & Majnon Jahromi, 2014*; J. Skevington, 2019, unpublished data) (Fig. 12 shows Middle Eastern distribution only).

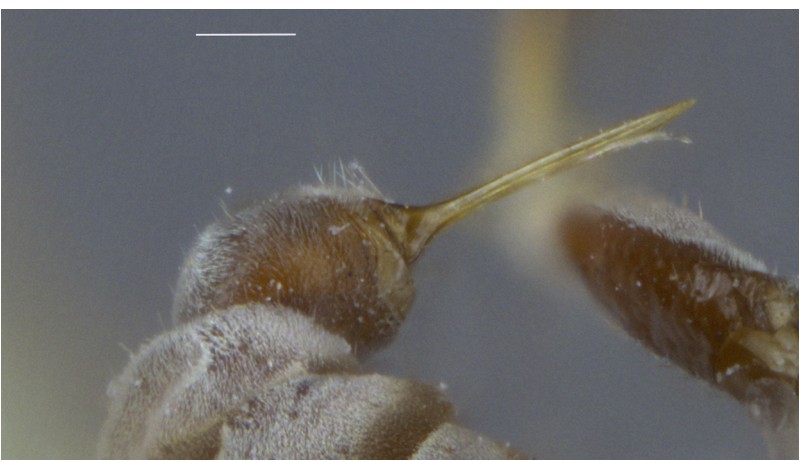

**Figure 13 Ovipositor of _Dasydorylas parazardouei_ Motamedinia and Skevington sp. nov. in lateral view.** Scale bar = 0.1 mm.

## Molecular variation

Based on uncorrected pairwise genetic distances (p-distance), this species is close to _D. gradus_ differing by 9.9–10.8% (Table 2).

## _Dasydorylas parazardouei_ Motamedinia & Skevington sp. nov.

Figures 1A, 1B, 10A–10E and 13

urn:lsid:zoobank.org:act:17ECF386-9A41-43F6-9B20-42EEF0E01600

## Materials examined

United Arab Emirates: holotype: male, Wadi Wurayah, 25.2400, 56.1700, 25.III.2007, sweep net, leg. F. Menzel & A. Stark, CNCD175301, GB: MN520761, CNC.
Paratypes: one male, one female, same data as holotype, CNCD175302, CNCD175303, GB: MN520771, CNC.

## Diagnosis

Abdomen brown (Fig. 1A); frons with a median keel in upper half; tergite 1 with three strong lateral bristles, arranged in one row; hypandrium distinctly bulging in underside (Fig. 10B); phallic guide strong, bow-like bent towards surstyli with two spines on each side (Figs. 10D and 10E).

## Description
### Male

Body length. 3.1–3.5 mm (excluding antennae). **Head.** Face dark, silver-gray pollinose. Frons brown, with a median keel in upper half. Pedicel brown with three short upper bristles and one long lower bristle; flagellum brown, short tapering (LF:WF = 2.1–2.2); arista brown. Eyes meeting for seventeen facets (Fig. 1B). Vertex and occiput brown. Pleura, prescutum, scutum and scutellum brown. Pleura gray pollinose. Postpronotal lobe pale, gray pollinose and with 2–3 postpronotal setae along upper margin. Scutum gray

pollinose, with patches of supra-alar setae. Scutellum gray pollinose, with a fringe of up to 10 dark setae. Subscutellum gray pollinose, Wing length: 3.1–3.3 mm. LW: MWW = 3.0–3.1. Wing almost entirely covered with microtrichia. Pterostigma brown and complete (LS:LTC = 1.0, LTC:LFC = 1.1). Halter length: 0.5 mm. Base dark, stem narrowly white and knob brown. **Legs.** All femora bearing two rows of dark anteroventral spines on apical one third. Tibiae with three rows of setae on anterior and posterior side, without apical spines. Hind tibia with some weak wrinkles midanteriorly. Tarsi brown and paler than tibiae, pulvilli smaller than distitarsi. **Abdomen.** Ground color brown. Tergite 1 gray pollinose with three strong lateral bristles, arranged in one row. Tergites 1–5 with brown setae. Syntergosternite 8 brown, without dorsal depression on side of right surstylus, as long as high (LS8:HS8 = 1.0). Membranous area vertically directed, broader in upper half, occupying about a third of the width of syntergosternite 8. Epandrium dark brown, pollinose except left edge. **Genitalia.** Genital capsule in dorsal view: surstyli brown, narrowly pale at apices, brown pollinose and rather symmetrical. Both surstyli with a blocky base and a broad finger-like projection at its apical inner corner, bent outward distally by 90°, base of right surstylus slightly wider than left surstylus. Genital capsule in lateral view: both surstyli in basal half broad, in apical half narrowed to form a finger-like process, which is bent towards the sternite by 90° (Figs. 10D and 10E). Phallus straight and slender, with one or two ejaculatory ducts, phallic guide strong, bow-like bent towards surstyli (Figs. 10D and 10E), with two dorsolateral spines at the end of basal half on either side (Figs. 10D and 10E). Ejaculatory apodeme funnel-shaped (Fig. 10C). Genital capsule in ventral view: gonopods minute and symmetrical, with elongated regions of distinctly stronger sclerotization (Fig. 10B).

### Female

Scape dark, with one upper short bristle. Pedicel with two short upper bristles and one long lower bristle. Flagellum short tapering. LF:WF = 2.0–2.2. Eyes separated. Frons dark, lower half silver-gray pollinose, otherwise shining. Frons anterior to ocellar triangle with median keels narrowing in lower half and ending in a tubercle shortly before antenna. Lateral rows of setae starting a bit before ocellar triangle and reaching down almost to tubercle. Postpronotal lobe yellow, gray pollinose with some light brown bristles. Pleura, prescutum, scutum and scutellum dark, gray pollinose. Femora with two small ventral rows of dark peg-like spine, restricted to apical one thirds. Tergites 1–5 gray pollinose laterally, extending onto dorsal surface along posterior margin. Tergites 2–5 with brown scattered bristles. Ovipositor light brown with some gray pollinosity, base nearly rounded; piercer straight, longer than base. LP:LB = 1.5., LDP:LPP = 3.3 (Fig. 13).

### Etymology

From para (=near) in Greek, referring to closely related to *Das. zardouei* (*Motamedinia et al., 2017a*).

### Distribution

United Arab Emirates (Fig. 12).

**Notes**

This species is related to *D. zardouei* *Motamedinia et al., 2017a*, *D. holosericeus* (*Becker, 1897*), re-described by *Kehlmaier (2005a)*, *D. roseri* (*Becker, 1897*), re-described by *Kehlmaier (2005a)*, *D. evanidus* (*Hardy, 1949*), re-described by *Földvári (2013)*, *D. orientalis* (*Koizumi, 1959*), re-described by *Kapoor, Grewal & Sharma (1987)* and also to *D. antennalis* (*Kapoor, Grewal & Sharma, 1987*) from southern India. The male of *D. parazardouei* differs from these species by the shape of the surstylus in lateral view and a semicircular bulge-shape of the hypandrial apodeme in ventral view (Fig. 10B).

**Molecular variation**

We have sequence data for male and female specimens of this species that show they are conspecific (0.00% uncorrected pairwise intraspecific difference—see Table 2). Based on uncorrected pairwise genetic distances (p-distance), this species is close to *D. gradus*, differing by 9.9% (Table 2).

***Dasydorylas zardouei*** *Motamedinia et al., 2017a*
Figures 11A–11E

**Materials examined**

Iran: one male, Kermanshah, Dodan, 35.000, 46.200, 1011m, 22.VII.2016, leg. M. Zardouei, Malaise trap, JSS52209, CNC.

**Diagnosis**

Abdomen dark; tergite 1 with four to five strong lateral bristles, arranged in one row; both surstyli with a blocky base and a broad finger-like projection at its apical inner corner, bent outward distally by 90° (Figs. 11D and 11E); base of right surstylus slightly wider than left surstylus in dorsal view (Fig. 11A); phallic guide with two spines on each side (Figs. 11D and 11E).

**Distribution**

Iran (*Motamedinia et al., 2017a*; J. Skevington, 2019, unpublished data) (Fig. 12).

**Notes**

This species is closely related to *D. holosericeus* (*Becker, 1897*) and *D. roseri* (*Becker, 1897*), both redescribed by *Kehlmaier (2005a)*, the Afrotropical *D. evanidus* (*Hardy, 1949*), redescribed by *Földvári (2013)*, the Oriental *D. orientalis* (*Koizumi, 1959*), redescribed by *Kapoor, Grewal & Sharma (1987)*, the southern Indian *D. antennalis* (*Kapoor, Grewal & Sharma, 1987*), described by *Kapoor, Grewal & Sharma (1987)* and *D. parazardouei* Motamedinia & Skevington from the United Arab Emirates. The males of *D. zardouei* differ from those of the other species by a different shape of gonopods in ventral view and shape of surstyli in lateral view.

## DISCUSSION

Sexes are dimorphic and difficult to associate in Pipunculidae, so it is now routine to use DNA barcodes to associate sexes (*Skevington, Kehlmaier & Ståhls, 2007*;

*Motamedinia et al., 2017a*; *Motamedinia, Skevington & Kelso, 2019*). *Dasydorylas derafshani*, which was described from the male (*Motamedinia et al., 2017a*), is associated here for the first time with *D. discoidalis*, which was known only from females (*Becker, 1897*). Interspecific genetic distances within the Middle Eastern *Dasydorylas* range from 6.9% (*D. discoidalis* to *D. dactylos*) to 16.8% (*D. parazardouei* to *D. discoidalis*), while intraspecific genetic distances range from 0% (within both *D. discoidalis* and *D. parazardouei* from the United Arab Emirates) to 3% (*D. discoidalis* from the United Arab Emirates and Iran). Based on uncorrected pairwise genetic distances (p-distance), *D. dactylos* is close to *D. discoidalis*, differing by 8.6% and *D. gradus* is most similar to *D. parazardouei* differing by 9.9% (Table 2).

# CONCLUSIONS

Prior to this study, the genus *Dasydorylas* included 32 worldwide species (J. Skevington, 2019, unpublished data), with only four, *D. discoidalis*, *D. gradus*, *D. horridus*, *D. zardouei* present in the Middle East. In this study we have extended the knowledge of this genus and described three new species, *D. dactylos* sp. nov., *D. forcipus* sp. nov., *D. parazardouei* sp. nov. and synonymized *D. derafshani* with *D. discoidalis*.

# ACKNOWLEDGEMENTS

We are grateful to C. Kehlmaier, A. Freidberg, N. Dorchin and A. van Harten for the loan of specimens from Israel and the United Arab Emirates to the CNC. Special thanks to E. Rakhshani from Zabol University for his ongoing collaboration and support. We are indebted to M. Parchami-Araghi, E. Gilasian, M. Zardouei, M. Ghaforimoghadam, K. Ghahari and H. Derafshan for collecting and providing the specimens from Iran.

## Funding

This work was supported by the Agriculture and Agri-Food Canada A-base grant to Jeffrey Hunter Skevington. The funders had no role in study design, data collection and analysis, decision to publish, or preparation of the manuscript.

## Grant Disclosures

The following grant information was disclosed by the authors:
Agriculture and Agri-Food Canada A-base.

## Competing Interests

The authors declare that they have no competing interests.

## Author Contributions

- Behnam Motamedinia conceived and designed the experiments, performed the experiments, analyzed the data, prepared figures and/or tables, authored or reviewed drafts of the paper, and approved the final draft.

- Jeffrey H. Skevington conceived and designed the experiments, authored or reviewed drafts of the paper, and approved the final draft.
- Scott Kelso conceived and designed the experiments, analyzed the data, authored or reviewed drafts of the paper, and approved the final draft.

### Data Availability

The sequences are available at GenBank: MN520769, MN520770, MN520764, MN520762, MN520767, MN520768, MN520763, MN520765, MN520766, MN520761, MN520771.

### New Species Registration

The following information was supplied regarding the registration of a newly described species:

Publication LSID: urn:lsid:zoobank.org:pub:A19B5B2E-817F-463C-A217-869C44C25C0A

*Dasydorylas dactylos* Motamedinia & Skevington sp. nov. LSID: urn:lsid:zoobank.org:act:BCE6B5FC-5C25-49C4-8473-F6D662DCE8CF

*Dasydorylas forcipus* Motamedinia & Skevington sp. nov. LSID: urn:lsid:zoobank.org:act:0F5D0097-4983-41FB-B35E-E71AB00C56C0

*Dasydorylas parazardouei* Motamedinia & Skevington sp. nov. LSID: urn:lsid:zoobank.org:act:17ECF386-9A41-43F6-9B20-42EEF0E01600.

### Supplemental Information

Supplemental information for this article can be found online at http://dx.doi.org/10.7717/peerj.8511#supplemental-information.

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
