# Peer review of "Taxonomic revision of *Dasydorylas* Skevington, 2001 (Diptera, Pipunculidae) in the Middle East"

_PeerJ, doi:10.7717/peerj.8511_

## Round 0.1 · original submission · Minor Revisions

This is a very nicely written and illustrated manuscript. Both reviewers were enthusiastic, generally noting a few typos and other relatively minor issues. One reviewer noted that the illustrations need to be inserted “into appropriate part of the key.” I am not really sure what they meant by this (?) because you do reference figures.

A few general points from me. 1) The details in the COI barcoding methods are overly exhaustive – I would reduce this section – you essentially used standard techniques and primers; 2) the barcoding aspect of the study is poorly integrated, provided as only notes in the taxonomic description. I am little surprised to not see a tree or other analyses. I suspect that you would enhance the extensibility of your paper by running a few more analyses and explicitly discussing the integrative aspects of your species delimitation; 3) how are species defined (empirically and/or conceptually)? You approach is seemingly morphological but clearly there’s some aspect of the molecular data that played a role. As noted above, discussing the interplay of these two data points may make the paper more interesting to others; 4) the text regarding ICZN rules needs a hard return and separate subheading rather than running together with the COI stuff. Also, ICZN registration numbers for each new species should be noted in their description to preclude any ambiguity about their status.

Reviewer 1 ·

Basic reporting

Illustrations need to be inserted into appropriate part of key, the failure to do this waste so much time usually having to re-edit keys to make them practical.

Experimental design

na

Validity of the findings

ok

Additional comments

Because highlighting lost when pasted here copy uploaded

32 Dorilas and Eudorylas. Dasydorylas was discovered during a comprehensive phylogenetic study
33 of world Eudorylini published by Skevington & Yeates in 2001.
This does not make sense, I think “coined” or first used better.

notopleuron often
39 with dense bush of long setae, femora often with posterdorsal row of long and back setae
Not apparent in any of the three species available to me? Perhaps these are small hairs rather than long setae??

sternite 3–5 with posterior
41 setae
Can’t see any in my specimens

41 setae, syntergosternite 8 with membranous area of normal size
Normal for what? In pipunculidae as a whole this feature ranges from absent to occupying whole width of syntergite. Try median size.

An identification key to Palaearctic and Afrotropical
50 species was provided by Kehlmaier (2005a) and Földvári (2013).
Kehlmaier (2005a) only covers European species, change this to
Either
An identification key to Palaearctic and Afrotropical
50 species was provided by Kehlmaier (2005a&b) and Földvári (2013).
OR
An identification key to European species was provided by Kehlmaier (2005a) and Afrotropical species by Földvári (2013).

70 For several species, only the male genitalia provide characters for accurate species identification
Seems presumptuous I would prefer - 70 For several species characters for accurate identification have only been found in the male genitalia.

postpronotal lobe with 4‒7 long setae
157 along upper margin;
Not apparent in my specimens, is this the same feature referred to in line 39 above pertaining there to notopleuron?

Annotated reviews are not available for download in order to protect the identity of reviewers who chose to remain anonymous.

Reviewer 2 ·

Basic reporting

The language is precise, describes morphological features and species relationships very in a detailed and clear manner.
Literature is used appropriately with references even to the distributional data, which is very helpful.
All figures and the two tables are informative and necessary, they give good support to the statements and species distances.

Experimental design

no comment

Validity of the findings

New species are well described with terms generally used in the field and the relevant reasoning is given with related species mentioned and COI genetic distances given.

Additional comments

A well-written account of the insect group with all the necessary ingredients and supporting information. The photographs provide all the required information and because they are photographs and not drawings they represent the actual structures (not representations of them), and are therefore very useful and advantageous.
Please find attached the few typos found in the text.
-the name Kehlmaier is misspelled 3 times
-table 2. has a value that should be bold

Annotated reviews are not available for download in order to protect the identity of reviewers who chose to remain anonymous.

---

## Round 0.2 · accepted · Accept

Absolutely outstanding job addressing the comments from both me and the reviewers. I particularly appreciated your attempts at dealing with the barcode data and also thought that your explanation of how your species were defined conceptually was nicely explored. It's a well written and wonderfully illustrated work!